# Right ventricular pressure–volume relations and effects of selective vena cava occlusion during cardiopulmonary resuscitation

Woo Jin Jung[1,2], Young-Il Roh[1,2], Soyeong Kim[1,2], Hyeonyoung Im[1,2], Yujin Lee[1,2], Sooyoung Sung[1,2], Jae Hun Han[3], Kyoung-Chul Cha[1,2☯*], Sung Oh Hwang[1,2☯*]

1 Department of Emergency Medicine, Yonsei University Wonju College of Medicine, Wonju, Republic of Korea, 2 Research Institute of Resuscitation Science, Yonsei University Wonju College of Medicine, Wonju, Republic of Korea, 3 Department of Biostatistics Science, Yonsei University Wonju College of Medicine, Wonju, Republic of Korea

☯ These authors contributed equally to this work.
* shwang@yonsei.ac.kr (SOH); chaemp@yonsei.ac.kr (KCC)

## Abstract

### Objectives

To investigate right ventricular hemodynamics and determine how vena cava occlusion (VCO) influences right ventricular pressure–volume dynamics during cardiopulmonary resuscitation (CPR).

### Methods

We used a swine model of electrically induced ventricular fibrillation. Five animals were allocated to Experiment I to observe right ventricular hemodynamics during standard CPR, and 25 were assigned to Experiment II to assess right ventricular hemodynamic changes following VCO. In Experiment I, all animals received standard CPR. In Experiment II, all animals received CPR but were randomized according to the protocols used, which differed by the order of interventions. The interventions included alternating superior VCO, inferior VCO, and no VCO. Hemodynamic parameters were measured during CPR, and corresponding right ventricular pressure–volume loops were generated.

### Results

Experiment I revealed that, during CPR, the right ventricular pressure–volume loop became trapezoidal in shape, with a progressive reduction in right ventricular volumes, including end-systolic volume (p = 0.029) and end-diastolic volume (p = 0.035). Experiment II demonstrated that systolic arterial pressure and end-tidal $CO_2$ levels were significantly lower during both superior and inferior VCO CPR than during no-VCO CPR (both p < 0.001). During superior and inferior VCO CPR, both end-systolic and end-diastolic right ventricular pressures were also significantly lower

**Data availability statement:** All relevant data are within the paper.

**Funding:** This work was supported by a National Research Foundation of Korea grant from the Korea Government Ministry of Science and Information & Communications Technology (ICT) (NRF-2020R1C1C1014376).

**Competing interests:** The authors have declared that no competing interests exist.

than those during no-VCO CPR (p < 0.001 and p = 0.003, respectively). Right ventricular stroke volumes did not differ significantly across the three VCO conditions. The shapes and values of right ventricular pressure–volume loops were relatively similar across the three VCO conditions.

## Conclusions

During CPR, the right ventricular pressure–volume loop transforms into a trapezoidal shape, resembling a left-leaning triangle, because the isovolumetric phases disappear. VCO during CPR reduces ventricular systolic pressures, indicating that a reduction in the venous return affects perfusion pressures during resuscitation.

## Introduction

For more than 60 years, chest compressions, relying solely on the manual force of rescuers, have been the primary method used for generating artificial circulation during cardiac arrest. However, the blood flow generated by chest compressions during cardiopulmonary resuscitation (CPR) reaches only 25–33% of the normal cardiac output, which is inadequate for maintaining optimal perfusion during cardiac arrest [1,2]. Although various mechanical devices and alternative techniques have been introduced to augment circulation during CPR, few have shown consistent improvement in the clinical outcomes of patients with cardiac arrest [3–6].

A better understanding of hemodynamic changes during CPR may reveal strategies for increasing circulatory effectiveness. Under normal physiological conditions, cardiac output is primarily determined by the preload, cardiac contractility, afterload, and heart rate. During cardiac arrest, the sudden loss of systolic and diastolic heart function results in circulatory collapse, with accompanying changes in the preload and afterload [7]. When CPR begins, the driving force for circulation shifts from cardiac contraction to chest compressions. This introduces a circulatory physiology that is fundamentally different compared to that involved during spontaneous circulation. Since active ventricular relaxation and atrial contraction do not occur during CPR, cardiac filling is likely influenced by extracardiac factors, such as chest wall recoil, venous return, and the passive mechanical properties of the ventricles [8,9]. To date, research has largely focused on the compression phase, with theories suggesting that blood flow is driven by direct external compression of the cardiac chambers or by increased intrathoracic pressure [10,11]. In contrast, the physiology of cardiac filling during the relaxation phase of CPR remains underexplored.

In this study, we investigated right ventricular (RV) hemodynamics during CPR and assessed how vena cava occlusion (VCO) influences RV pressure–volume (PV) dynamics during this treatment.

## Materials and methods

The data supporting the findings of the study are available from the corresponding authors upon reasonable request.

## Ethical considerations

This study was approved by the Institutional Animal Care and Use Committee of Yonsei University Wonju College of Medicine, Wonju, Republic of Korea (approval number: YWC-200305–2; approval date: 05/12/2021). All study procedures adhered to the tenets of the Declaration of Helsinki.

## Study design and animals

The study was composed of two experiments: Experiment I involved observing RV hemodynamics during standard CPR, whereas Experiment II entailed evaluating the changes in RV hemodynamics after VCO. We used the swine cardiac arrest model with electrically induced ventricular fibrillation (VF). In total, 30 male Yorkshire pigs (35−45 kg) were included. We allocated 5 animals to Experiment I and 25 animals to Experiment II.

## Animal model preparation

**Anesthesia and ventilation.**  The pigs were provided free access to food and water up to 12 h before the experiment. Anesthesia was induced with intramuscular ketamine (15 mg/kg) and xylazine (2 mg/kg) and was maintained by inhalation of 3% isoflurane. After sedation, the pigs were placed in a prone position and endotracheally intubated using a 7.5-Fr cuffed tube. Subsequently, they were repositioned to the supine position, and volume-controlled ventilation via an animal ventilator (Fabius GS, Drager Medical Inc., Telford, PA, USA) was started. Ventilation was initiated with a tidal volume of 10−15 mL/kg, a positive end-expiratory pressure of 3−5 cmH$_2$O, and a respiratory rate of 12−20 breaths per minute, and was adjusted to maintain end-tidal carbon dioxide (ETCO$_2$) levels between 35 and 40 mmHg under continuous CO$_2$ monitoring (CO$_2$SMO Plus; Novametrix Medical Systems, Wallingford, CT, USA). Continuous electrocardiography, using lead II, was performed throughout the procedure.

**Catheterization.**  For all animals, surgical drapes were applied, and surgical sites were sterilized using betadine solution. The left or right femoral artery was cannulated using a 5.5-Fr introducer sheath catheter (Arrow International Inc., Reading, PA, USA) according to the Seldinger method. A 5-Fr micromanometer-tipped catheter (Millar Instruments, Inc., Houston, TX, USA) was inserted via the femoral artery for continuous aortic pressure recording. An introducer sheath catheter was inserted into the right external jugular vein, and a PV catheter (5-Fr Millar Mikro-tip pressure–volume catheter; Millar Instruments, Inc.) was introduced into the RV to measure the pressure and composite volume. The right internal carotid artery was exposed and a vascular flowmeter (Transonic Systems Inc., Ithaca, NY, USA) was placed for continuous measurement of carotid blood flow (CBF). An introducer catheter was inserted into the right internal jugular vein and an electrode (5-Fr bipolar lead; Arrow International Inc.) for inducing VF was inserted into the RV.

In the animals assigned to Experiment II, the left external jugular vein and the left or right femoral vein were cannulated using 5.5-Fr introducer catheters according to the Seldinger method. Balloon catheters (5 Fr-balloon embolectomy catheters; LeMaitre Vascular, Inc., Burlington, MA, USA) were inserted into the superior vena cava (SVC) and inferior vena cava (IVC) for applying VCO. To confirm the proper placement of the balloon catheters, 2 mL of normal saline mixed with contrast dye (XENETIX 300®, Guerbet, France) was injected into the balloon. The final position of the catheters was confirmed by taking a portable plain chest X-ray (RHT, Inc., Gwangju, Republic of Korea) (S1 Fig). Subsequently, 100 IU/kg of heparin was injected intravenously to prevent thrombus formation. In all cases, arterial blood gas was analyzed, and troponin I levels measured. The animals were stabilized for 10 min after completion of the procedure.

**Induction of VF.**  After catheterization and baseline measurements were completed, VF was induced by passing alternating current (60 Hz, 30 mA) into the RV through a pacing electrode by using an electrical stimulation device for 10–20 s. Induction of VF was confirmed by the presence of the typical VF electrocardiogram waveform and loss of aortic pulse pressure.

## Experimental procedures

**Experiment I.** After 2 min of VF, chest compressions with a depth of 5 cm were performed at a rate of 100 compressions/min by using a mechanical CPR device (LUCAS-2® Chest Compression System; Physio-Control, Redmond, WA, USA). Positive-pressure ventilation using room air at a tidal volume of approximately 300 mL was delivered by using a resuscitator bag (Silicone resuscitator 87005133; Laerdal Medical, Stavanger, Norway). Two ventilations were provided per every 30 compressions. CPR was performed for 26 min, and no attempts at restoring spontaneous circulation, such as by defibrillation or epinephrine administration, were made during CPR.

**Experiment II.** After 2 min of VF, chest compressions with a depth of 5 cm were performed at a rate of 100 compressions/min by using a mechanical CPR device (LUCAS-2® Chest Compression System; Physio-Control). Positive-pressure ventilation with room air at a tidal volume of approximately 300 mL was delivered by using a resuscitator bag (Silicone resuscitator 87005133; Laerdal Medical). Two ventilations were provided for every 30 compressions.

Hemodynamic effects were evaluated during CPR by alternately occluding and releasing the vena cava for 2-min periods. The sequence of SVC occlusion (SVCO) and IVC occlusion (IVCO) was alternated in the protocols to minimize time-related fluctuations in hemodynamic parameters. The protocol sequences were No-VCO–SVCO–No-VCO–IVCO (2 min each) in Protocol 1 and No-VCO–IVCO–No-VCO–SVCO (2 min each) in Protocol 2. The sequence for each protocol was repeated three times (S2 Fig). Randomization for protocol allocation was performed by using sealed envelopes containing the assigned protocol type; these envelopes were opened by the principal investigator immediately before VF induction. VCO was achieved by filling the balloon of the catheter inserted into the SVC or IVC with 3 mL of normal saline to maintain the balloon diameter at approximately 1 cm. To deflate the balloon to its original state, the saline solution in the balloon was removed. All animals in Experiment II received CPR for approximately 26 min. During CPR, no attempt was made to restore spontaneous circulation, such as by using defibrillation or administering epinephrine.

## Outcomes

The hemodynamic parameters measured included arterial pressures, CBF, RV pressures (RVPs), and RV volumes (RVVs). These measured values were used to create PV loops. Additional RV hemodynamic parameters, including stroke volume (SV), cardiac output, stroke work, the derivative of pressure over time (dP/dt), the derivative of volume over time (dV/dt), and the time constant of RV relaxation (Tau), were calculated. Hemodynamic parameters were measured at 2, 4, 12, 20, and 26 min after CPR initiation. The measured or calculated values were averaged over the 2-min CPR cycle, with exclusion of the first 15 s during which the CPR method was switched.

## Sample size calculation

The sample size was determined with reference to the limited relevant literature available. The sample size for Experiment I was set at five pigs, referring to previous studies that successfully performed PV loop measurements using five animals [12]. Based on two pilot experiments for Experiment II, we extracted preliminary data regarding the difference in RVVs. The sample size was calculated based on the difference in RVV measured at 2 min after the SVCO or IVCO period compared to that measured during standard CPR. Assuming that at least two values were secured for each sector and at least six sample RVVs were extracted per one experiment, the sufficient sample size was determined by determining the difference in the overall RVV mean and standard deviation. A sample size of 25 could achieve 70% power to detect a difference of −3.7 between the null hypothesis mean of zero and the alternative hypothesis mean of 3.7, with a known standard deviation of 18.2 and with a significance level (alpha) of 0.05 based on a two-sided, one-sample $t$-test. We also set the sample size referencing an animal study that assessed left ventricular end-systolic PV relations with transient IVCO by using an impedance catheter in 24 animals [13].

## Statistical analysis

Continuous variables are summarized as mean values ± standard deviation and were compared using analysis of variance or the Kruskal−Wallis test, as appropriate. Categorical data are summarized as frequencies and percentages and were compared using the chi-square or Fisher's exact test, as appropriate. Repeated-measures analysis of variance was employed to compare hemodynamic parameter values across different time points, and was also used to compare hemodynamic parameter values over time and across interventions at different intervention points in each group. Statistical significance was defined as a p-value less than 0.05. Statistical analyses were performed using SAS (version 9.4; SAS Institute Inc., Cary, NC, USA).

## Results

### Baseline characteristics

Among the animals included in Experiment II, 13 and 12 animals received CPR according to Protocols 1 and 2, respectively. The baseline characteristics of the animals, including body weight, baseline laboratory values, and hemodynamic parameters, are shown in the Supporting Information (S1 Table).

### RV hemodynamics during spontaneous circulation and CPR

In Experiment I, changes in RV hemodynamics during CPR were observed. Compared to the PV loop during spontaneous circulation, the PV loops during CPR became smaller over time and became trapezoidal in shape, due to loss of isovolumetric contraction and relaxation (Fig 1). The PV loops at 2 and 4 min after CPR initiation became smaller and triangular due to loss of isovolumetric contraction and relaxation. A noticeable shift towards lower pressures and volumes indicated impaired RV performance early after CPR initiation. At 12 and 20 min after CPR initiation, the loops continued to shrink, with a further decline in pressure and volume, suggesting progressive hemodynamic deterioration and reduced cardiac output. At 26 min after CPR initiation, the PV loop was the smallest, showing minimal pressure and volume changes, indicating severely compromised RV function.

A gradual decrease in systolic, diastolic, and mean arterial pressures and CBF was observed during CPR, although it was not statistically significant. RV end-systolic pressure (RVPes) and RV end-diastolic pressure (RVPed) remained unchanged (p = 0.887 for RVPes, p = 0.056 for RVPed), whereas RV end-systolic volume (RVVes) and RV end-diastolic volume (RVVed) decreased over time during CPR (p = 0.029 for RVVes and p = 0.035 for RVVed). The change in RV SV was not statistically significant (p = 0.197) (Table 1; Fig 2).

### Effects of VCO during CPR on hemodynamics

Experiment II revealed that systolic arterial pressure and $ETCO_2$ levels were lower during SVCO and IVCO than during No-VCO conditions (both p < 0.001). Mean and diastolic arterial pressures and CBF were similar during the No-VCO, SVCO, and IVCO conditions. RVPes and RVPed were significantly lower during SVCO and IVCO than during No-VCO (p < 0.001 for RVPes and p = 0.003 for RVPed), whereas RVVs and SV did not differ significantly across conditions (p = 0.414 for RVVes, p = 0.850 for RVVed, and p = 0.841 for SV; Fig 3; Table 2). The overall PV loop shapes and values did not differ significantly among SVCO, IVCO, and No-VCO (Fig 4, S3 Fig). The loop eccentricity declined over time, reflecting that the loop became relatively wider horizontally (S4 Fig). The end-systolic pressure–volume relationship slope, reflecting RV contractility, gradually decreased over the course of CPR in all three groups, with the lowest value occurring in the period of spontaneous circulation (S5 Fig). The end-diastolic pressure–volume relationship slope, reflecting ventricular compliance, remained low overall and showed only modest changes over time and across groups (S2 Table, S6 Fig).

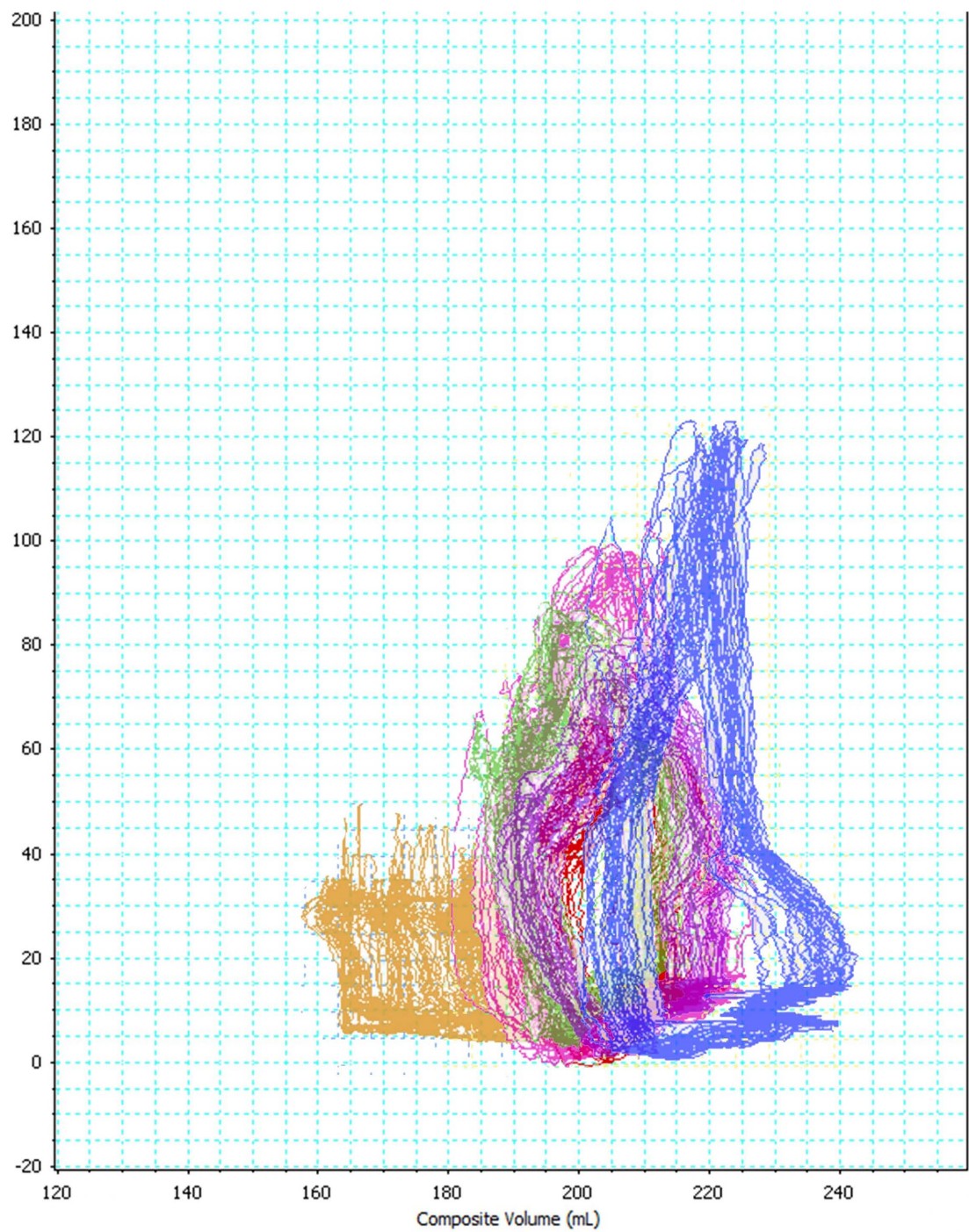

**Fig 1. Representative RV PV loops during spontaneous circulation and during CPR.** The x-axis displays the composite RVV (mL), and the y-axis shows the RVP (mmHg). The yellow loop, indicating spontaneous circulation, has a typical elliptical shape. The measurement loops at 2, 4, 12, 20, and 26 min after CPR initiation are shown in blue, green, pink, purple, and red, respectively. During CPR, the PV loops progressively take on a trapezoidal shape, indicating the loss of isovolumetric contraction and relaxation. This highlights the passive conduit role of the RV under external chest compressions. Unlike the elliptical loop during spontaneous circulation, these trapezoidal loops exhibit a limited volume range and show a more linear pressure increase during compression, with sharp drops during decompression. These patterns suggest modifications in RV filling and ejection mechanics during CPR. RV, right ventricular; PV, pressure–volume; CPR, cardiopulmonary resuscitation; RVV, right ventricular volume; RVP, right ventricular pressure.

**Table 1. Hemodynamic measurement at baseline and during CPR in the No-VCO group.**

| | Spontaneous circulation | Time elapsed after CPR initiation | | | | | p-value |
|---|---|---|---|---|---|---|---|
| | | 2 min | 4 min | 12 min | 20 min | 26 min | |
| Systolic arterial pressure, mmHg | 108.2 ± 15.3 | 81.4 ± 7.9 | 81.4 ± 10.0 | 79.3 ± 12.6 | 65.3 ± 21.1 | 55.6 ± 22.3 | 0.236 |
| Diastolic arterial pressure, mmHg | 72.7 ± 16.6 | 23.0 ± 2.7 | 19.9 ± 4.4 | 22.0 ± 7.1 | 14.8 ± 9.0 | 13.7 ± 8.7 | 0.290 |
| Mean arterial pressure, mmHg | 84.5 ± 13.0 | 42.5 ± 3.2 | 40.4 ± 4.8 | 41.1 ± 8.4 | 31.6 ± 7.8 | 28.1 ± 10.4 | 0.106 |
| PR, bpm | 75.9 ± 58.0 | 101.8 ± 0.1 | 101.8 ± 0.3 | 101.9 ± 0.2 | 101.7 ± 0.2 | 101.8 ± 0.1 | 0.622 |
| CBF, mL/min | 365.6 ± 65.3 | 117.6 ± 42.3 | 96.5 ± 30.9 | 73.2 ± 38.8 | 59.5 ± 26.9 | 49.3 ± 5.5 | 0.232 |
| ETCO$_2$, mmHg | 27.8 ± 3.3 | 18.1 ± 4.4 | 17.7 ± 5.2 | 18.4 ± 4.3 | 18.6 ± 5.0 | 19.1 ± 8.9 | 0.952 |
| RVPes, mmHg | 31.4 ± 6.7 | 109.9 ± 37.4 | 135 ± 63.8 | 156.5 ± 83.5 | 139.8 ± 69.8 | 137.7 ± 54.5 | 0.887 |
| RVPed, mmHg | 9.9 ± 4.4 | 15.7 ± 2.9 | 17.1 ± 2.2 | 19.8 ± 3.6 | 15.2 ± 1.9 | 15.4 ± 1.4 | 0.056 |
| RVVes, mL | 168.0 ± 29.8 | 192.1 ± 26.6 | 182.2 ± 27.1 | 172.4 ± 25.3 | 167.1 ± 33.3 | 170.2 ± 26.8 | 0.029 |
| RVVed, mL | 197.4 ± 32.6 | 218.9 ± 30.3 | 206.6 ± 32.0 | 196.1 ± 31.8 | 188.3 ± 39.0 | 191.0 ± 32.3 | 0.035 |
| SV, mL | 42.9 ± 1 7.7 | 40 ± 12.5 | 34.8 ± 14.9 | 33.2 ± 13.5 | 31.0 ± 11.1 | 29.3 ± 10.5 | 0.197 |

CPR, cardiopulmonary resuscitation; VCO, vena cava occlusion; PR, pulse rate; CBF, carotid blood flow; ETCO$_2$, end-tidal carbon dioxide; RVPes, end-systolic right ventricular pressure; RVPed, end-diastolic right ventricular pressure; RVVes, end-systolic right ventricular volume; RVVed, end-diastolic right ventricular volume; SV, right ventricular stroke volume

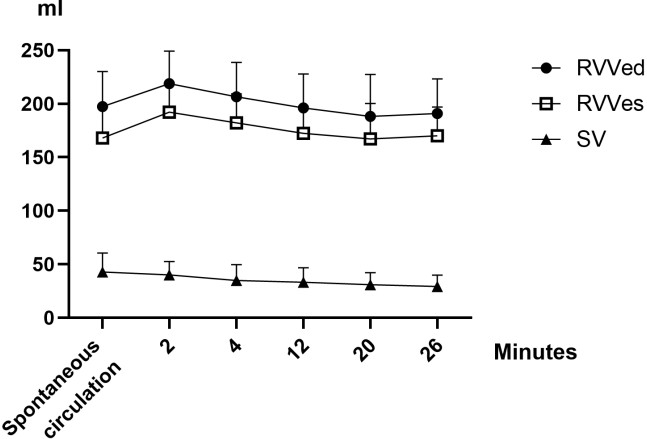

**Fig 2. Changes in RVVs during spontaneous circulation and during CPR.** The right ventricular systolic and diastolic volumes initially increase following the induction of VF and subsequently gradually decrease during the course of CPR.

SVCO, IVCO, and No-VCO conditions did not differ significantly in terms of the hemodynamic parameters according to the time elapsed from CPR initiation; these included CBF, ETCO$_2$, RVPes, RVPed, RVVes, RVVed, and SV. However, a decline in systolic blood pressure was noted (p =0.236 for No-VCO, p =0.002 for SVCO, and p =0.008 for IVCO; S3 Table).

RVVs, right ventricular volumes; RVVed, end-diastolic right ventricular volume; RVVes, end-systolic right ventricular volume; SV, right ventricular stroke volume; VCO, vena cava occlusion; SVCO, superior vena cava occlusion; IVCO, inferior vena cava occlusion

RV PV, right ventricular pressure–volume; CPR, cardiopulmonary resuscitation; VCO, vena cava occlusion; SVCO, superior vena cava occlusion; IVCO, inferior vena cava occlusion; SV, right ventricular stroke volume; RVP, right ventricular pressure; RVV, right ventricular volume.

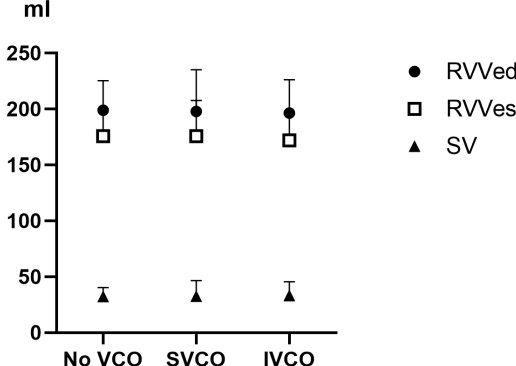

**Fig 3. Comparison of RVVs according to the intervention type.** No significant differences were observed in RVVed and RVVes, or in the SV between the No-VCO, SVCO, and IVCO conditions.

**Table 2. Comparisons of hemodynamic values between groups.**

|  | No-VCO CPR | SVCO CPR | IVCO CPR | p-value |
|---|---|---|---|---|
| Systolic arterial pressure, mmHg | 72.5±17.5 | 58.7±20.8 | 55.4±22.5 | <0.001 |
| Diastolic arterial pressure, mmHg | 19.0±7.5 | 18.2±11.4 | 19.6±14.9 | 0.805 |
| Mean arterial pressure, mmHg | 38.3±8.8 | 32.9±11.2 | 33.1±16.1 | 0.141 |
| PR, bpm | 102.1±1.3 | 104.8±25.0 | 102.5±5.9 | 0.606 |
| CBF, mL/min | 73.5±33.5 | 88.2±101.0 | 85.3±106.4 | 0.777 |
| ETCO$_2$, mmHg | 18.2±5.0 | 12.1±5.5 | 11.2±5.2 | <0.001 |
| RVPes, mmHg | 135.8±60.4 | 108.0±51.6 | 95.8±51.9 | 0.003 |
| RVPed, mmHg | 17.6±3.3 | 12.9±3.6 | 10.1±5.4 | <0.001 |
| RVVes, mL | 175.8±24.5 | 175.7±32.0 | 171.9±23.1 | 0.414 |
| RVVed, mL | 198.8±26.6 | 197.8±37.4 | 196.3±29.8 | 0.850 |
| SV, mL | 32.3±8.1 | 32.7±13.9 | 33.3±12.3 | 0.841 |

CPR, cardiopulmonary resuscitation; VCO, vena cava occlusion; SVCO, superior vena cava occlusion; IVCO, inferior vena cava occlusion; PR, pulse rate; CBF, carotid blood flow; ETCO$_2$, end-tidal carbon dioxide; RVPes, end-systolic right ventricular pressure; RVPed, end-diastolic right ventricular pressure; RVVes, end-systolic right ventricular volume; RVVed, end-diastolic right ventricular volume; SV, right ventricular stroke volume

## Discussion

In this study, we investigated RV hemodynamics during CPR and evaluated the effects of VCO on RV PV dynamics during this treatment. In Experiment I, by observing the RV PV loop under spontaneous circulation and during CPR, we found that, as CPR progressed, the shape of the RV PV loop gradually changed to a shape that was significantly different from that of the loop obtained during spontaneous circulation. In Experiment II, we observed that VCO during CPR resulted in a decrease in both RV and systemic arterial pressures, without changing RVVs as compared to those of the non-occluded state. To the best of our knowledge, no previous study has analyzed the RV PV relationship during CPR in a swine model of cardiac arrest.

Our observations during Experiment I revealed that, during CPR, the RV PV loop took on a trapezoidal shape, which was distinct from the pattern observed during spontaneous circulation. This change could be attributed to the loss of both isovolumetric contraction and relaxation periods. Furthermore, this indicated a physiological change caused by pulmonary hypertension during CPR. Upon VF onset, arterial collapse leads to an abrupt decrease in afterload, while dysfunction of the papillary muscles and changes in ventricular geometry caused by CPR contribute to valvular insufficiency [7,14,15].

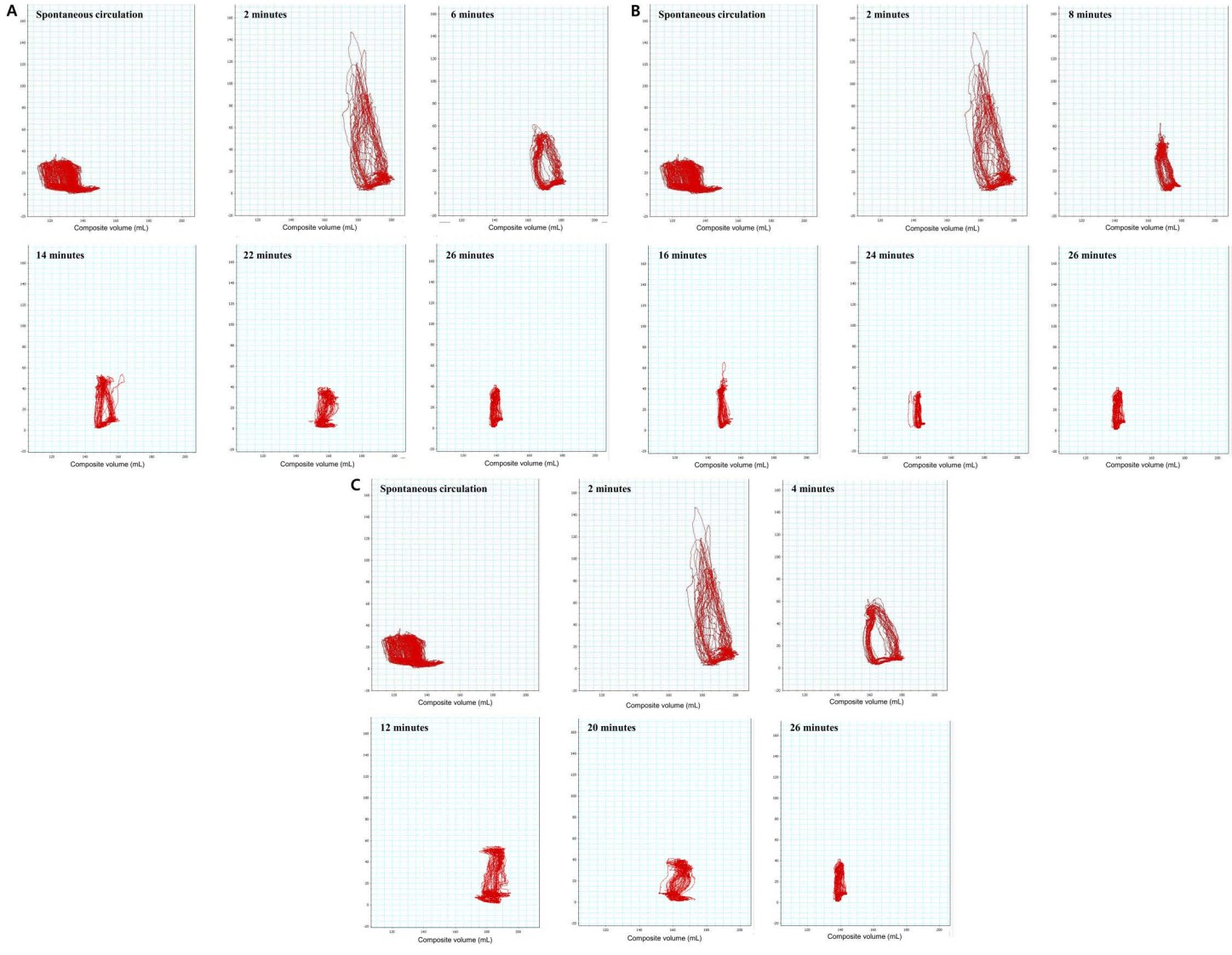

**Fig 4. Representative RV PV loops measured during spontaneous circulation and during CPR under different conditions.** No significant differences were observed in RV PV relationships during standard CPR without VCO, with SVCO, and with IVCO. (A) PV loops during standard CPR without VCO. The loops show compression-induced pressure fluctuations with reduced SV and a rightward shift as compared to those obtained during spontaneous circulation. (B) PV loops during CPR with SVCO. The overall shape and position of the loops are comparable to those observed during standard CPR without VCO, with no significant changes in SV or pressure development. (C) PV loops during CPR with IVCO. Similar to CPR with SVCO, the loops demonstrate no appreciable difference in RVP or RVV parameters as compared to standard CPR without VCO.

The transformation of the PV loop during CPR as compared to that during spontaneous circulation appears to result from a combination of arterial collapse, impaired ventricular relaxation, and valvular dysfunction, leading to the disappearance of the isovolumetric periods.

To date, research on the mechanisms of blood flow generated during CPR has primarily focused on the compression phase, while the mechanisms underlying diastolic filling remain largely unknown. Two main factors that determine diastolic dynamics are relaxation and compliance [16,17]. Relaxation is an ATP-dependent, energy-consuming process; thus, in energy-deficient ischemic conditions, relaxation is delayed [18]. Compliance is determined by chamber distensibility. In a

study using domestic pigs in which VF was induced, prolonged CPR resulted in increased LV wall thickness, leading to a "stone heart" condition, which was associated with progressive reductions in diastolic and stroke volumes, ultimately causing diastolic dysfunction [19]. The increase in myocardial stiffness due to the "stone heart" phenomenon leads decreased compliance. We also observed a gradual decrease in RVVs with the progression in CPR. This finding suggests that prolonged myocardial ischemia after VF results in a progressive loss of ventricular compliance. The reduction in ventricular volume due to the progressive loss of compliance may contribute to the gradual decline in cardiac output observed during CPR [20].

In Experiment II, we assessed the impact of occlusion of the SVC and the IVC on RV filling by using selective VCO. When either the SVC or IVC was occluded, no changes in RVV parameters, including RVVed, RVVes, and SV, were noted. However, systolic arterial pressure, RVPes, RVPed, and ETCO$_2$ decreased. Additionally, the hemodynamic effects of SVCO and IVCO were similar. In a previous animal study, acute IVCO for 5 min decreased LV systolic and diastolic pressures, LV volume, cardiac output, and systemic blood pressure, whereas SVCO reduced LV diastolic pressure and volume without affecting cardiac output or systemic blood pressure [21]. Another animal study demonstrated a progressive reduction in systemic blood pressure and a leftward and downward shift of end-systolic PV and pressure-dimension points during IVCO [13]. In both healthy individuals and in patients with congestive heart failure, IVCO lasting 10 − 15 min led to a decrease in RV filling pressure, mean pulmonary artery pressure, systolic and diastolic arterial pressures, and cardiac output [22]. In patients with congestive heart failure, SVCO for 5 or 10 min resulted in reductions in right atrial pressure, pulmonary artery pressure, and pulmonary capillary wedge pressure, without affecting arterial pressure [23]. Unlike our study, which investigated the effects of VCO during CPR, these previous studies examined its impact under conditions of spontaneous circulation. Therefore, their findings have limitations when applied to an analysis of our results. Unlike during spontaneous circulation, the heart acts as a vascular conduit during VF-related cardiac arrest, and does not undergo intrinsic contraction and relaxation. During CPR, blood flow is generated by the force exerted on the chest during the compression phase, while cardiac filling occurs during the relaxation phase due to the pressure gradient between venous pressure and RV pressure, or due to the difference between extrathoracic and intrathoracic pressures [24]. Due to the fundamental physiological differences between spontaneous and CPR-induced circulation, the hemodynamic changes caused by VCO during spontaneous circulation cannot be directly extrapolated to CPR physiology. Nevertheless, our observations demonstrated that VCO during CPR leads to hemodynamic changes, including reduced pulmonary and systemic arterial pressures, as observed during spontaneous circulation. However, our study could not fully explain why RVVs remained unchanged despite reductions in systolic arterial pressure and RVPs following VCO. During CPR, venous return may occur from other venous compartments or collateral pathways, while the pulmonary compartment and right atrium may act as volume buffers [25,26]. Consequently, significant changes in RVVs might not be apparent. The present study demonstrated a tendency of progressive reduction in the RVVs and SV with an increasing CPR duration. This observation suggested that venous return progressively declines during ongoing resuscitation. Such hemodynamic deterioration may, at least in part, account for the decreased likelihood of successful resuscitation with prolonged CPR. Interventions aimed at augmenting venous return, such as timely administration of intravenous fluid, may mitigate the decline in SV and improve the likelihood of successful resuscitation. Future studies, particularly in clinical settings, are warranted to determine the optimal fluid strategy during prolonged resuscitation and to assess its impact on both short- and long-term outcomes.

Our study had some limitations. Since this was an animal study, the experimental conditions may differ from those extant during human cardiac arrest. We used a model with only basic life support, using a mechanical CPR device, which differs from advanced cardiovascular life support available for humans. Unlike during spontaneous circulation, the RV is compressed more than the LV by chest compressions during CPR, because it is located closer to the sternum [25]. This may have influenced the measurements from the PV catheter. The duration of VCO may have been insufficient, resulting in minimal hemodynamic effects. In this study, the duration of VCO was based on a previous animal study, which demonstrated that 10–15 s of IVCO reduced LV systolic pressure by approximately 60 mmHg, when evaluating

LV PV relationships [13]. Considering that blood flow during CPR is decreased to about one-third to one-quarter of that during normal circulation, we extended the occlusion duration to 2 min [20]. As this study was designed to investigate RV hemodynamics, we did not assess the effects of VCO on outcomes such as restoration of spontaneous circulation or survival. Further research is warranted to evaluate the impact of VCO on these clinical outcomes. Finally, further mechanistic exploration, including preload surrogates, echocardiography, or estimates of vascular resistance, would facilitate an understanding of the hemodynamics of venous return during CPR. Future studies are needed to assess the hemodynamic impact of longer periods of VCO and to introduce tools capable of real-time preload/afterload tracking. However, as precise hemodynamic measurements are difficult to obtain during CPR, new methodology beyond PV loop measurements will be necessary.

## Conclusions

This study showed that, during CPR, the RV PV loop transforms into a trapezoidal shape, resembling a leftward-leaning triangle, due to the disappearance of the isovolumetric contraction and relaxation phases. VCO during CPR reduced ventricular systolic pressures, indicating that a reduced venous return affects perfusion pressures during resuscitation. Our study contributes to an understanding of CPR physiology and optimization of resuscitation strategies. The unique transformation of the RV PV loop highlights the need to reconsider the current understanding of the hemodynamics involved during CPR. Given that VCO reduces arterial pressures but not RV volumes, optimizing venous return may play a critical role in maintaining CPR-generated blood flow. Strategies such as improved chest compression techniques or adjunctive mechanical support may help to enhance venous return and improve perfusion.

## Supporting information

**S1 Fig. Confirmation of catheter placement.** Chest anteroposterior X-ray image showing catheter placement. The image was obtained using a portable chest X-ray device. The red arrow indicates a PV catheter in the right ventricle. The blue arrows indicate the balloon catheters in the proximal portion of the SVC and IVC. PV, pressure–volume; SVC, superior vena cava; IVC, inferior vena cava.
(DOCX)

**S2 Fig. Protocols for Experiment II.** The experimental protocols used during CPR to assess the hemodynamic effects of VCO. Two protocols were applied to minimize time-dependent variations. In Protocol 1, 2-min periods of CPR with No-VCO, SVCO, No-VCO, and IVCO were sequentially performed. In Protocol 2, the sequence was No-VCO, IVCO, No-VCO, and SVCO, for 2 min each. The sequence was repeated three times. VF, ventricular fibrillation; S-CPR, standard cardiopulmonary resuscitation with no vena cava occlusion; S-CPR + SVCO, standard cardiopulmonary resuscitation with superior vena cava occlusion, S-CPR + IVCO, standard cardiopulmonary resuscitation with inferior vena cava occlusion.
(TIF)

**S3 Fig. RV PV loops during CPR within the first 8 (A), 16 (B), and 24 min (C) in a representative case.** Measurement loops for the No-VCO, SVCO, and IVCO periods are shown in blue, green, and red, respectively. Compared to the spontaneous circulation period, marked in black, when CPR is performed with No-VCO, both RVVed and RVVes increase, but the effective volume does not seem to increase significantly. In addition, no significant differences were observed in the RV stroke volume between the No-VCO, SVCO, and IVCO conditions. RV PV, right ventricular pressure–volume; CPR, cardiopulmonary resuscitation; VCO, vena cava occlusion; SVCO, superior vena cava occlusion; IVCO, inferior vena cava occlusion; RVVed, end-diastolic right ventricular volume; RVVes, end-systolic right ventricular volume.
(DOCX)

**S4 Fig. RV PV loop eccentricity during CPR.** Measurement loops for the No-VCO, SVCO, and IVCO periods are shown in yellow, blue, and green, respectively. Loop eccentricity was calculated as (RVPes - RVPed)/ (RVVed - RVVes), where higher values indicates a more elongated loop, reflecting changes in RV loading conditions or systolic/diastolic imbalance. The eccentricity decreases over time, reflecting that the loop became relatively wider horizontally (greater volumetric than pressure fluctuations). RV PV, right ventricular pressure–volume; VCO, vena cava occlusion; SVCO, superior vena cava occlusion; IVCO, inferior vena cava occlusion; SV, right ventricular stroke volume; Pedv, developed pressure (maximum pressure minus minimal pressure); Spont; spontaneous circulation period.
(TIF)

**S5 Fig. End-systolic pressure–volume loop relationship slope during CPR.** ESPVR was calculated as RVPes/ (RVVes - $V_0$) and was used as an index of RV contractility. The ESPVR slope gradually decreased (decreased contractility) over the course of CPR in all three groups, with the lowest value occurring in the period of spontaneous circulation. ESPVR, end-systolic pressure–volume relationship; VCO, vena cava occlusion; SVCO, superior vena cava occlusion; IVCO, inferior vena cava occlusion; RVPes, end-systolic right ventricular pressure; ESV, end-systolic right ventricular volume; Spont; spontaneous circulation period.
(TIF)

**S6 Fig. End-diastolic pressure–volume loop relationship slope during CPR.** EDPVR was defined as RVPed/ RVVed and was used as an index of diastolic stiffness, with higher values indicating reduced compliance. The EDPVR slope remains low overall and shows only modest changes over time and across groups. EDPVR, end-diastolic pressure–volume relationship; VCO, vena cava occlusion; SVCO, superior vena cava occlusion; IVCO, inferior vena cava occlusion; RVPed, end-diastolic right ventricular pressure; EDV, end-diastolic right ventricular volume; Spont; spontaneous circulation period.
(TIF)

**S1 Table. Baseline characteristics and measurements of animals included in Experiments I and II.**
(DOCX)

**S2 Table. Comparison of RV PV loop characteristics between interventions according to the time elapsed from CPR initiation.**
(DOCX)

**S3 Table. Comparison of hemodynamic measurements between interventions according to the time elapsed from initiation of CPR.**
(DOCX)

## Acknowledgments

The authors thank Ms. Min Gyeong Choi and Mr. Myeong Ha Kim for their technical support.

## Author contributions

**Conceptualization:** Woo Jin Jung, Kyoung-Chul Cha, Sung Oh Hwang.

**Data curation:** Sooyoung Sung, Jae Hun Han.

**Investigation:** Hyeonyoung Im, Yujin Lee.

**Methodology:** Woo Jin Jung.

**Software:** Young-Il Roh, Soyeong Kim.

**Supervision:** Kyoung-Chul Cha, Sung Oh Hwang.

**Visualization:** Sooyoung Sung.

**Writing – original draft:** Woo Jin Jung.

**Writing – review & editing:** Kyoung-Chul Cha, Sung Oh Hwang.

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
