## [Decision Letter · Decision Letter 0]

2 Jun 2025

Dear Dr. Oh Hwang,

Thank you for submitting your manuscript to PLOS ONE. After careful consideration, we feel that it has merit but does not fully meet PLOS ONE’s publication criteria as it currently stands. Therefore, we invite you to submit a revised version of the manuscript that addresses the points raised during the review process.

We look forward to receiving your revised manuscript.

Kind regards,

Ahmet Çağlar, Associate Professor

Academic Editor

PLOS ONE

Journal Requirements:

“This work was supported by a National Research Foundation of Korea grant from the Korea Government Ministry of Science and Information & Communications Technology (ICT) (NRF-2020R1C1C1014376)”

Reviewers' comments:

Reviewer's Responses to Questions

**Comments to the Author**

1. Is the manuscript technically sound, and do the data support the conclusions?

Reviewer #1: No

Reviewer #2: Yes

2. Has the statistical analysis been performed appropriately and rigorously?

Reviewer #1: Yes

Reviewer #2: Yes

3. Have the authors made all data underlying the findings in their manuscript fully available?

Reviewer #1: Yes

Reviewer #2: Yes

4. Is the manuscript presented in an intelligible fashion and written in standard English?

Reviewer #1: Yes

Reviewer #2: Yes

Reviewer #1: The study was done to explore right ventricular hemodynamic and assess the impact of vena cava occlusion (VCO) on RV pressure-volume during cardiopulmonary resuscitation (CPR). The significance was to see what occurs during relaxation phase of cardiac arrest when vena cava occlusion was applied. It showed that right ventricular end systolic and diastolic significantly decreased with no change in right ventricular volume. This concluded that if venous return is reduced during CPR, then perfusion pressure is affected.

Regarding their conclusions, it is well known that one of the causes of cardiac arrest is hypovolemia where there is decrease in venous return. This is managed by giving intravenous fluids and continuing CPR until return of spontaneous circulation is achieved.

The sample size is not large enough as they used 30 pigs with 5 pigs had no vena cava occlusion, but standard CPR and 25 pigs had venal cava occlusion during CPR. The study did not show if pigs with vena cava occlusion return to spontaneous circulation faster than pigs that did not have vena cava occlusion. If the randomization number was equal, and the study looked at the effect of vena cava occlusion in improving CPR and return of spontaneous circulation during ventricular fibrillation, it will be more beneficial to help improve patient care.

The study did not focus on what happened to the ventricular fibrillation when the occlusion was performed and thus it did not show any advantage of performing this procedure during tachyarrhythmias. This should be a major focus as new advances is needed to help bring the patients back to spontaneous circulation.

The study did look at the right ventricular pressure volume during the relaxation phase during cardiac arrest and what happens if vena cava occlusion was applied. I don’t see that they have found new data that will improve CPR during cardiac arrest but only minor physiological changes that did not affect those pigs who were in ventricular fibrillation and had CPR performed on them.

When I read the article, I was excited to see if the procedure had any influence the arrhythmia. What happened to the pigs that have the occlusion? Did they survive? Did they have a better outcome than the pigs with only standard CPR performed on them. These questions were on my mind, but they were not mentioned. The focus was on the right ventricle and the article concluded that perfusion pressure is affected if venous return is reduced which is obvious and well-known fact.

Reviewer #2: The No-VCO group includes only 5 animals, which limits the statistical power for temporal comparisons during standard CPR. This weakens confidence in findings related to time-dependent RV hemodynamic deterioration.

VCO periods of only 2 minutes may be too brief to fully manifest hemodynamic effects, as acknowledged by the authors themselves. The rationale for this duration should be discussed in more detail, perhaps informed by prior literature or pilot data.

A key finding—that RV volumes did not significantly change despite reduced RV and systemic pressures during VCO—is counterintuitive. The authors speculate about decreased compliance or short VCO duration, but this remains unresolved. More mechanistic insight or additional measurements (e.g., preload indicators, ventricular elastance) would strengthen this point.

While the study includes several figures and tables, Figure 1–4 and supplemental data are essential to interpreting the results, yet only briefly described. Including more figure-specific interpretation in the main text would help readers grasp the findings.

Terms like “trapezoidal PV loop” could be better defined or quantified (e.g., loop area, width/height ratio). Figures should also include loop overlays for easier visual comparison.

This study provides important insights into RV mechanics during CPR and could significantly advance understanding in the field. However, the issues with group size balance, unexplained physiological findings, and underdeveloped interpretation limit its current impact. Addressing these concerns—particularly the unexpected RV volume results and the limited duration of VCO—would considerably enhance the paper's value.

**Do you want your identity to be public for this peer review?** For information about this choice, including consent withdrawal, please see our Privacy Policy

Reviewer #1: **Yes: ** Ali Abdullah Ashkanani

Reviewer #2: No

---

## [Author Response · Author response to Decision Letter 1]

15 Jul 2025

The manuscript has been rechecked and the necessary changes have been made in accordance with the reviewers’ suggestions. The responses to all comments have been prepared and attached separately. Revisions in the manuscript are indicated with red fonts. Please check the response file to the reviewers' comments.

Thank you.

---

## [Decision Letter · Decision Letter 1]

20 Jul 2025

Dear Dr. Oh Hwang,

We look forward to receiving your revised manuscript.

Kind regards,

Ahmet Çağlar, Associate Professor

Academic Editor

PLOS ONE

Journal Requirements:

**Additional Editor Comments:**

Dear Author;

This manuscript is still needing a major revision. After revisions made, will be considered again.

Your sincerely.

Reviewers' comments:

Reviewer's Responses to Questions

**Comments to the Author**

Reviewer #2: (No Response)

2. Is the manuscript technically sound, and do the data support the conclusions?

Reviewer #2: Yes

3. Has the statistical analysis been performed appropriately and rigorously?

Reviewer #2: Yes

4. Have the authors made all data underlying the findings in their manuscript fully available?

Reviewer #2: Yes

5. Is the manuscript presented in an intelligible fashion and written in standard English?

Reviewer #2: Yes

Reviewer #2: This manuscript presents a well-executed experimental study evaluating right ventricular (RV) hemodynamics and the impact of superior and inferior vena cava occlusion (VCO) during cardiopulmonary resuscitation (CPR) using a swine model. The topic is relevant and underexplored, especially in regard to diastolic physiology and RV pressure-volume (PV) dynamics during cardiac arrest. The study introduces novel insights into RV mechanics under CPR conditions and may lay the groundwork for improved resuscitation strategies. However, there are several critical issues regarding design rationale, statistical power, mechanistic interpretation, and clinical applicability that should be addressed or acknowledged more thoroughly.

COMMENTS

Consider discussing more directly how the observed hemodynamic changes could theoretically affect CPR success or guide future therapeutic approaches.

Include a clearer justification of sample size based on power calculations, variability estimates, or effect size assumptions.

This is a key physiological paradox. A deeper mechanistic exploration—possibly through preload surrogates, echocardiography, or vascular resistance estimates—would enhance the scientific contribution.

Acknowledge this limitation more explicitly, and suggest longer VCO durations or real-time preload/afterload tracking in future studies.

Include a table summarizing key geometric PV loop characteristics or define shape transformation metrics quantitatively. Loop overlays for visual comparison would enhance clarity.

Use consistent definitions of acronyms (e.g., RVVes, RVVed) throughout text and figures to reduce reader confusion.

The manuscript is well written, with high technical clarity. However, some repetitions exist (e.g., "trapezoidal loop resembling a leftward-leaning triangle" appears in several sections).

This study is a meaningful step toward better understanding of RV physiology during CPR, particularly in relation to venous return. However, its translational potential is currently limited by the absence of outcome-based endpoints, small control group size, and unresolved hemodynamic findings. With appropriate clarification, extended mechanistic insight, and strengthened figure interpretation, this work could serve as a valuable foundation for future resuscitation research.

**Do you want your identity to be public for this peer review?** For information about this choice, including consent withdrawal, please see our Privacy Policy

Reviewer #2: No

---

## [Author Response · Author response to Decision Letter 2]

2 Sep 2025

We thank to the editor and reviewers for their thoughtful recommendations for improving the quality of our manuscript. (Manuscript No. PONE-D-25-22980: Right ventricular pressure-volume relations and effect of selective vena cava occlusion during cardiopulmonary resuscitation) We have revised the manuscript based on the comments and suggestions of the reviewer. We have also provided point-by-point responses to the reviewer’s comments, and the changes made according to the reviewer’s suggestions are shown in red highlights in the enclosed revised manuscript. Please check the attached file.

---

## [Decision Letter · Decision Letter 2]

10 Sep 2025

Right ventricular pressure-volume relations and effects of selective vena cava occlusion during cardiopulmonary resuscitation

PONE-D-25-22980R2

Dear Dr. Oh Hwang,

We’re pleased to inform you that your manuscript has been judged scientifically suitable for publication and will be formally accepted for publication once it meets all outstanding technical requirements.

Kind regards,

Ahmet Çağlar, Associate Professor

Academic Editor

PLOS ONE

Reviewers' comments:

Reviewer's Responses to Questions

**Comments to the Author**

Reviewer #2: All comments have been addressed

2. Is the manuscript technically sound, and do the data support the conclusions?

Reviewer #2: Yes

3. Has the statistical analysis been performed appropriately and rigorously?

Reviewer #2: I Don't Know

4. Have the authors made all data underlying the findings in their manuscript fully available?

Reviewer #2: Yes

5. Is the manuscript presented in an intelligible fashion and written in standard English?

Reviewer #2: Yes

Reviewer #2: can be aaccepted in current form. THANK YOU FOR YOUR EFFORT IN REVISION

OF PONE-D-25-22980R2, entitled "Right ventricular pressure-volume relations and effects of selective vena cava occlusion during cardiopulmonary resuscitation".

**Do you want your identity to be public for this peer review?** For information about this choice, including consent withdrawal, please see our Privacy Policy

Reviewer #2: **Yes: ** EJDER SAYLAV BORA

---

## [Editor Report · Acceptance letter]

PONE-D-25-22980R2

PLOS ONE

Dear Dr. Oh Hwang,

I'm pleased to inform you that your manuscript has been deemed suitable for publication in PLOS ONE. Congratulations! Your manuscript is now being handed over to our production team.

Kind regards,

on behalf of

Dr. Ahmet Çağlar

Academic Editor

PLOS ONE